# A review of recent and future marine extinctions

Pablo del Monte-Luna[1] , Miguel Nakamura[2], Alba Vicente[1,3], Lilian B. Pérez-Sosa[2], Arturo Yáñez-Arenas[1], Andrew W. Trites[4] and Salvador E. Lluch-Cota[5]

[1]Departamento de Pesquerías y Biología Marina, Instituto Politécnico Nacional (IPN), La Paz, Mexico; [2]Departamento de Probabilidad y Estadística, Centro de Investigación en Matemáticas A.C. (CIMAT), Guanajuato, Mexico; [3]Departament de Dinàmica de la Terra i de l'Oceà, Facultat de Ciències de la Terra, Universitat de Barcelona-UB, Barcelona Spain; [4]Institute for the Oceans and Fisheries, University of British Columbia, Vancouver, BC, Canada and [5]Programa de Ecología Pesquera, Centro de Investigaciones Biológicas del Noroeste, La Paz, Mexico

## Review

**Cite this article:** del Monte-Luna P, Nakamura M, Vicente A, Pérez-Sosa LB, Yáñez-Arenas A, Trites AW and Lluch-Cota SE (2023). A review of recent and future marine extinctions. *Cambridge Prisms: Extinction*, **1**, e13, 1–9 https://doi.org/10.1017/ext.2023.11

**Keywords:**
Biodiversity loss; sixth mass extinction; overfishing; extirpation; extinction risk

**Corresponding author:**
Pablo del Monte-Luna;
Email: pdelmontel@ipn.mx

## Abstract

Between 20 and 24 marine extinctions, ranging from algal to mammal species, have occurred over the past 500 years. These relatively low numbers question whether the sixth mass extinction that is underway on land is also occurring in the ocean. There is, however, increasing evidence of worldwide losses of marine populations that may foretell a wave of oncoming marine extinctions. A review of current methods being used to determine the loss of biodiversity from the world's oceans reveals the need to develop and apply new assessment methodologies that incorporate standardized metrics that allow comparisons to be made among different regions and taxonomic groups, and between current extinctions and past mass extinction events. Such efforts will contribute to a better understanding of extinction risk facing marine flora and fauna, as well as the ways in which it can be mitigated.

## Impact statement

The number of global marine extinctions that have been reported during the past 500 years is less than 25 species. However, the rapid loss of marine populations around the world due mostly to human activities may soon trigger more extinctions that imperil marine ecosystems and the basic goods and services they provide humanity. However, uncertainties remain in detecting the disappearance of marine species and populations that can be addressed using new extinction metrics and methodologies to define conservation reference points and contextualize the current loss of marine biodiversity.

## Introduction

Extinctions are a hot topic in ecology, as are concerns about the cumulative effects that the worldwide extinctions of thousands of species are having on human societies (Cardinale et al., 2012). In the terrestrial realm, extinctions are occurring at a speed and magnitude comparable to mass extinctions in the distant past (Barnosky et al., 2011). However, this same diagnosis is not as equally conclusive in marine ecosystems. Hence, this review 1) provides an overview of modern marine extinctions; 2) addresses key concepts that underlie the current biodiversity crisis in the world's oceans; and 3) identifies priorities in the study of marine extinctions.

In this review, we define extinction as the disappearance of the last individual of a species (International Union for the Conservation of Nature (IUCN), 2019), and define extirpation or as the disappearance of a population (Powles et al., 2000). Neo-extinctions and extirpations refer to events that occurred since 1500 (MacPhee and Flemming, 1999), an era of traceable worldwide environmental conditions. We address both extinctions and extirpations because modern marine extinctions appear to be relatively rare, and the bulk of available information on biodiversity loss in the sea concerns extirpations.

## Extinction rates of modern marine species and populations

The argument that life on Earth is experiencing a sixth mass extinction is based on the observation that the speed at which terrestrial species have disappeared due to human actions after five centuries is far greater than the background extinction rate in the distant past (also known as the expected or normal extinction rate; Benton, 2003). The background extinction rate for fossil marine invertebrates ranges between 0.1 and 1 extinctions per million extant species per year (E/MSY; Pimm et al., 1995) and is as high as 1.8 E/MSY for some megafaunal groups (Dirzo and Raven, 2003; Barnosky et al., 2011; Proença and Pereira, 2013). In contrast, the current extinction rate of the terrestrial biota stands between 0.1 and 100 E/MSY (Pimm et al., 2006;

Millennium Ecosystem Assessment, 2005) and up to 150–260 E/MSY (150,000–260,000 extinctions in the past 500 years, out of 2 million known terrestrial species; Cowie et al., 2022), although this upper limit could be overestimated (Stork, 2010; Briggs, 2017). Despite the considerable variance of these estimates, there is a consensus that the current extinction rate in terrestrial species is, on average, well above background extinction rates. This fundamental piece of evidence supports the idea of an ongoing sixth mass extinction.

No explicit estimates of the current extinction rate of marine species have been published so far, although Briggs (2017) notes that they are extremely small. The estimated number of extant marine species is 1.8–2 million (Mora et al., 2011), of which 36 species were thought to have gone extinct during the past half millennium – but later lowered to just 20–24 species after reexamination of the available data (Table 1). Thus, the current extinction rate of marine species (E/MSY based on 24 extinctions, 1.8 million extant species, and 500 years) is 0.03, while the background extinction rate of the marine biota in the geologic past is 0.1–1.0 (Millennium Ecosystem Assessment, 2005). However, the large uncertainty in the estimates associated with small sample sizes (20–24 cases of extinction) and limited sampling effort means that it is just as plausible that the extinction rate has remained constant during the past 500 years as that it has changed (Nakamura et al., 2013).

At the population level, loss rates in marine populations are now 2–10 times higher than they were 500 years ago (Dulvy et al., 2003, 2009; del Monte-Luna et al., 2007; Nakamura et al., 2013). This accelerated loss of marine biodiversity is consistent with growing reports of marine defaunation being caused by human activities (Harnik et al., 2012; McCauley et al., 2015), and may mark the beginning of a sixth mass extinction in the oceans. Rather than waiting for marine species to be declared extinct, monitoring declining populations of species that are at risk may prove to be a timely means by which extinction risk can be assessed.

### The study of marine biodiversity loss in brief

Two of the first studies to cast doubt about the widely held presumption that marine life was resilient to extinctions concerned the neo-extinctions of four marine snails (Carlton, 1993; Vermeij, 1993; Table 1). Subsequent studies cast further doubt on this presumption, and questioned whether or not a new wave of extinctions was underway (Malakoff, 1997; Roberts and Hawkins, 1999). Adding further weight to this concern was a compiled list of 12 modern extinct marine species that included several imperiled species (Carlton et al., 1999; Powles et al., 2000). Collectively, these studies deemed marine ecosystems to be at equal risk of losing species as terrestrial ecosystems, but pointed to a lack of procedures to identify extinct marine organisms.

Historical overfishing of the world oceans, along with other synergistic threats, were shown to have significantly reduced the abundance of over 30 populations and were initially believed to have caused the extinction of at least two species (Jackson et al., 2001). A further 112 extirpations and 21 marine extinctions were later determined to have occurred during the past 500 years, along with identifying their proximal causes (mainly overexploitation and habitat modification; Dulvy et al., 2003, 2009). While there is some question of whether this number of extirpations is overestimated by a factor of two (del Monte-Luna et al., 2007), there

is general consensus on the number of marine neo-extinctions that have occurred.

During the past 500 years, marine ecosystems have experienced environmental stressors similar if not worse to those associated with pre-historic mass extinction events (e.g., climate change, ocean acidification, and sea-level rise) with a notable difference in the rapid pace with which environmental stressors are now occurring due to human activities (Harnik et al., 2012). These stressors are associated with marine extirpations and near-extirpations, and are leading to reduced cross-system connectivity, reduced genetic diversity, disrupted ecosystem stability, and altered biogeochemical cycles (McCauley et al., 2015). Most, if not all, studies concur that the loss of marine populations has increased worldwide during the last century, and that the number of documented extinctions has remained rather small.

It is unclear why so few marine extinctions have been reported if they have indeed occurred. One possibility is that marine extinctions are equally common as terrestrial extinctions, but are simply harder to detect (Webb and Mindel, 2015). Another possibility is that less research effort is directed at the most endangered marine species relative to the increasingly large number of studies on commercial species (game fish; Guy et al., 2021). Both possibilities point to the need to increase research efforts toward the most imperiled marine populations and species before they reach a point of no return.

Anthropogenic pressure can reduce populations to a point where they cannot fulfill their functional roles within ecosystems (McCauley et al., 2015). Such ecological extinctions have been documented in terrestrial (e.g., the empty forest; Redford, 1992) and marine ecosystems (e.g., the eradication of sea otters, sheepshead labrid fish, and spiny lobsters along the North American Pacific coast; Jackson et al., 2001; Jackson, 2008). These case studies show the particular harm that ecological extinctions have when the dwindling populations are structural ecological engineers (del Monte-Luna et al., 2007) that maintain corals and kelp forests, or are the primary providers of top-down control of trophic cascades and energy flow (Eger and Baum, 2020).

Less is known about the effects of marine extinctions on microbial biodiversity (i.e., eubacteria, archaea, protists, single-celled fungi, and viruses), which play important roles in ecosystem functioning. However, the co-dependence (and co-evolution) of microbes and their animal and plant hosts species suggests a high likelihood that losses of marine fishes and other species could result in co-extinctions of microbial life forms. There may also be some interplay between microbes and the biodiversity of pathogens that rely on bacterial associates. If so, the effects of climate change on ecological processes that depend on microbial communities may have secondary effects on extinction rates of marine species (Hunter-Cevera et al., 2005; Weinbauer and Rassoulzadegan, 2007; Thaler, 2021).

The fossil record reveals differences in the local and global variables that affected the likelihood of species surviving periods of background and mass extinction events. Most notably, variables such as planktotrophic larval development, broad geographic distributions (Payne and Fionnegan, 2007), high species richness (Jablonski, 1986), and small body sizes enhanced the survival of species and genus during background times (Payne et al., 2016). In contrast, survival during mass extinction events was enhanced for species and entire lineages that were geographically broadly dispersed (Jablonski, 1986) but was not influenced by body size (which was inversely but moderately or not at all associated with extinction probability; Payne et al., 2016). However, survival during the

**Table 1.** Declarations of extinction (E) and re-evaluation as not extinct (NE) for 36 marine species based on 18 assessments published from 1975 to 2022, as well as their current status.

| Taxonomic groups | 1975[1] | 1993[2] | 1993[3] | 1999[4] | 2000[5] | 2000[6] | 2001[7] | 2003[8] | 2005[9] | 2007[10] | 2007[11] | 2009[12] | 2013[13] | 2015[14] | 2015[15] | 2019[16] | 2020[17] | 2022[18] | Current status |
|---|---|---|---|---|---|---|---|---|---|---|---|---|---|---|---|---|---|---|---|
| **Kingdom: Animalia** **Phylum: Chordata** **Class: Mammalia** | | | | | | | | | | | | | | | | | | | |
| *Zalophus japonicus* | – | – | – | E | – | E | – | – | – | E | – | – | – | – | – | – | – | E | E |
| *Neomonachus tropicalis* | – | E | – | – | – | E | – | E | – | – | E | – | – | – | – | – | – | E | E |
| *Mustela macrodon* | – | E | – | – | – | E | – | E | – | – | E | – | – | – | – | – | – | E | E |
| *Hydrodamalis gigas* | – | E | – | E | – | E | – | E | – | – | E | – | – | – | – | – | – | E | E |
| **Class: Aves** | | | | | | | | | | | | | | | | | | | |
| *Tadorna cristata* | – | E | – | E | – | E | – | – | – | – | – | – | – | – | – | – | – | NE | NE |
| *Camptorhynchus labradorius* | – | E | – | E | – | E | – | E | – | – | E | – | – | – | – | – | – | E | E |
| *Mergus australis* | – | E | – | E | – | E | – | E | – | – | E | – | – | – | – | – | – | E | E |
| *Oceanodroma macrodactyla* | – | E | – | E | – | E | – | – | – | – | – | – | – | – | – | – | – | NE | NE |
| *Pterodroma rupinarum* | E | – | – | – | – | – | – | – | – | – | – | – | – | – | – | – | – | E | E |
| *Pterodroma imberi* | – | – | – | – | – | – | – | – | – | – | – | – | – | E | – | – | – | – | E |
| *Pterodroma caribbaea* | – | E | – | E | – | E | – | – | – | – | – | – | – | – | – | – | – | NE | NE |
| *Bulweria bifax* | E | – | – | – | – | – | – | – | – | – | – | – | – | – | – | – | – | E | E |
| *Phalacrocorax perspicillatus* | – | E | – | E | – | E | – | E | – | – | E | – | – | – | – | – | – | E | E |
| *Haematopus meadewaldoi* | – | E | – | E | – | E | – | E | – | – | E | – | – | – | – | – | – | E | E |
| *Pinguinus impennis* | – | E | – | E | – | E | – | E | – | – | E | – | – | – | – | – | – | E | E |
| **Class: Actinopterygii** | | | | | | | | | | | | | | | | | | | |
| *Psephurus gladius* | – | – | – | – | – | – | – | – | – | – | – | – | – | – | – | – | E | E | E |
| *Azurina eupalama* | – | – | – | – | – | – | – | E | – | – | NE | – | – | – | – | – | – | NE | NE |
| *Anampses viridis* | – | – | – | – | E | – | – | E | – | – | E | – | NE | – | – | – | – | NE | NE |
| *Prototroctes oxyrhynchus* | – | – | – | – | – | – | – | E | – | – | E | – | – | – | – | – | – | E | E |
| *Coregonus oxyrhynchus* | – | – | – | – | – | – | – | – | E | – | – | – | – | – | – | – | – | E | E* |
| *Sympterichthys unipennis* | – | – | – | – | – | – | – | – | – | – | – | – | – | – | – | – | – | E | E |
| **Class: Chondrichthyes** | | | | | | | | | | | | | | | | | | | |
| *Carcharhinus obsolerus* | – | – | – | – | – | – | – | – | – | – | – | – | – | – | – | E | – | NE | NE |
| **Phylum: Cnidaria** **Class: Hexacorallia** | | | | | | | | | | | | | | | | | | | |
| *Edwardsia ivelli* | – | – | – | E | – | – | – | E | – | – | NE | – | – | – | – | – | – | NE | NE |
| *Siderastrea glynni* | – | – | – | – | – | – | – | E | – | – | – | – | – | – | – | – | – | NE | NE |

(*Continued*)

**Table 1.** (*Continued*)

| Taxonomic groups | 1975[1] | 1993[2] | 1993[3] | 1999[4] | 2000[5] | 2000[6] | 2001[7] | 2003[8] | 2005[9] | 2007[10] | 2007[11] | 2009[12] | 2013[13] | 2015[14] | 2015[15] | 2019[16] | 2020[17] | 2022[18] | Current status |
|---|---|---|---|---|---|---|---|---|---|---|---|---|---|---|---|---|---|---|---|
| | | | | | | | | | | Year of publication | | | | | | | | | |
| Class: Hydrozoa | | | | | | | | | | | | | | | | | | | |
| *Millepora boschmai* | – | – | – | – | – | – | E | – | – | – | E | – | – | – | – | – | – | NE | NE |
| Phylum: Mollusca Class: Gastropoda | | | | | | | | | | | | | | | | | | | |
| *Lottia (Collisella) edmitchelli* | – | – | E | E | – | E | – | E | – | – | NE | – | – | – | – | – | – | E | E |
| *Lottia alveus* | – | – | E | E | – | E | – | E | – | – | E | – | – | – | – | – | – | NE | NE |
| *Cerithidea fuscata* | – | E | E | E | – | E | – | E | – | – | E | – | – | – | – | – | – | – | E |
| *Littoraria flammea* | – | – | E | E | – | E | – | E | – | – | E | – | – | – | NE | – | – | E | E |
| *Phyllaplysia smaragda* | – | – | – | E | – | – | – | – | – | – | – | – | – | – | – | – | – | – | E* |
| *Stiliger vossi* | – | – | – | E | – | – | – | – | – | – | – | – | – | – | – | – | – | – | E* |
| *Haliotis sorenseni* | – | – | – | – | – | – | – | E | – | – | NE | – | – | – | – | – | – | NE | NE |
| Class: Bivalvia | | | | | | | | | | | | | | | | | | | |
| *Pholadomya candida* | – | – | – | E | – | – | – | – | – | – | – | NE | – | – | – | – | – | – | NE |
| Phylum: Arthropoda Class: Crustacea | | | | | | | | | | | | | | | | | | | |
| *Sirenocyamus rhytinae* | – | – | – | E | – | – | – | – | – | – | – | – | – | – | – | – | – | – | E* |
| Phylum: Rhodophyta Class: Florideophyceae | | | | | | | | | | | | | | | | | | | |
| *Gigartina australis* | – | – | – | – | – | – | – | E | – | – | E | – | – | – | – | – | – | – | E |
| *Vanvoorstia bennettiana* | – | – | – | – | – | – | – | E | – | – | E | – | – | – | – | – | – | E | E |
| Total extinct (E) species | 2 | 12 | 4 | 19 | 1 | 16 | 1 | 20 | 1 | 1 | 16 | 0 | 0 | 1 | 0 | 1 | 1 | 18 | 24 |
| Total non-extinct (NE) species | 0 | 0 | 0 | 0 | 0 | 0 | 0 | 0 | 0 | 0 | 4 | 1 | 1 | 0 | 1 | 0 | 0 | 11 | 12 |
| Total non-evaluated (–) species | 34 | 24 | 32 | 17 | 35 | 20 | 35 | 16 | 35 | 35 | 16 | 35 | 35 | 35 | 35 | 35 | 35 | 7 | 0 |

*Note:* Species not assessed within a publication (–), and those that have been declared extinct but should continue to be evaluated (E*), are also shown.

*References*: 1. Olson (1975); 2. Vermeij (1993); 3. Carlton et al. (1993); 4. Carlton et al. (1999); 5. Hawkins et al. (2000); 6. Wolff (2000); 7. Glynn et al. (2001); 8. Dulvy et al. (2003); 9. Freyhof and Schöter (2005); 10. Sakahira and Niimi (2007); 11. del Monte-Luna et al. (2007); 12. Díaz et al. (2009); 13. Russell and Craig (2013); 14. Tennyson et al. (2015); 15. Dong et al. (2015); 16. White et al. (2019); 17. Zhang et al. (2020); 18. International Union for Conservation of Nature (IUCN) (2022).

*Species declared extinct that we consider should continue to be subject to evaluation.

Permian extinction event was enhanced for skeletal organisms that could contend with elevated carbon dioxide in their bloodstreams (hypercapnia), while nasal respiratory turbinates (that act as countercurrent heat exchangers during lung ventilation), together with burrowing behavior in vertebrates, were key to survival during the Triassic extinction event (Knoll et al., 2007).

The current crisis facing marine biodiversity is primarily affecting large species, including herbivores, and is disrupting trophic food webs (Payne et al., 2016; Atwood et al., 2020). Body size has been a good predictor of extinction risk in marine tetrapods (del Monte-Luna and Lluch-Belda, 2003), but is less effective among fishes and of no consequence for invertebrates (González-Valdovinos et al., 2019) where factors such as climate, habitat alteration and loss, and motility may determine current extinction proneness as occur with terrestrial organism (Munstermann et al., 2021). However, should marine extinctions be biased towards larger species of fishes and invertebrates, it would follow that tropical ecosystems may be most at risk due to having higher concentrations of human activities (Finnegan et al., 2015).

In addition to studies focused on the loss of marine biodiversity, there have been reports of extinctions – some of which may not be well supported (see non-extinct reports in Table 1). For instance, the periwinkle *Littoraria flammea* was first considered extinct in China by Carlton (1993) but later rediscovered and placed as a possible morphological variation of *L. melanostoma* (Dong et al., 2015). Similarly, a marine reef fish from Mauritius (*Anampses viridis*) was reported to be extinct (Hawkins et al., 2000; Dulvy et al., 2003; del Monte Luna et al., 2007), but is now considered to be the adult male color form of *A. caeruleopunctatus*, which is common and widespread throughout the Indo-West Pacific region. Another marine species whose extinction (Freyhof and Schöter, 2005) has been disputed over its taxonomic identity is the houting *Coregonus oxyrhynchus* from the North and Baltic Seas (Borcherding et al., 2010; Dierking et al., 2014). However, these few cases of questionable extinctions do not negate the fact that the number of documented cases of recent global marine extinctions has increased over time (Table 1).

The number of reported marine extinctions during the past 500 years stands between 20 and 24 species across all marine groups, from algae to mammals (Table 1). Such low numbers of extinctions casts doubt on the claim of an ongoing mass extinction in the oceans. However, 60–112 marine populations were lost during the years 1500–2000, and there is concern that the continued increase of anthropogenic stressors will eradicate other populations of threatened species such as sharks and rays (Dulvy et al., 2021; Pacoureau et al., 2021). The cumulative eradications of populations may also shorten the time between an initial perturbation and the extinction of a species (extinction debt; Figueiredo et al., 2019), making a rising of a wave of extinctions in the oceans an imminent reality (Rogers and Laffoley, 2013). Such irreversible "tipping points" in marine biodiversity will have undesirable consequences on basic goods and services that support human wellbeing (McCauley et al., 2010, 2015), as has been observed in terrestrial ecosystems (Dirzo et al., 2014).

## The pace of marine biodiversity loss

### Unraveling past extinctions to understand the present

Understanding past extinctions has been an important means to comprehend the causes and effects of current biotic crises. It is possible, for example, to use the taxonomic and geologic information compiled in paleontological databases to determine biotic richness, and extinction and origination rates – and to interpret how past ecosystems were formed and how they changed through time. Paleontological databases can also be used to derive predictors of extinction vulnerability of marine biota (Finnegan et al., 2015).

The fossil record shows that life on Earth evolved from the marine realm. Indeed, most of the fossil record is composed of marine organisms, which reflects their proneness to fossilization, their marked biodiversity, and their wide geographic distribution. In comparison, the fossil record of terrestrial species is relatively sparse due to fewer opportunities to be buried and fossilized. Thus, fossilized terrestrial species are generally scattered, composed of incomplete or fragmentary remains, and rarely show continental distributions. However, the terrestrial biota diversified rapidly following colonization of the land, and represents 85–95% of the current total biodiversity (Benton, 2016).

Studies of past extinctions have tended to rely on the richness of the marine fossil record to evaluate the effects of past biotic crises. In contrast, evaluations of current biotic crises tend to rely on species presence and absence from continental ecosystems, due in part to the availability of data. Using different data sources to compare past and current extinctions presents some challenges. First, the data come from different ecosystem-type biotas (marine vs. terrestrial). Second, different metrics are used to compare the mass extinctions that occurred over millions of years with the biotic crises that have occurred on a scale of hundreds of years (Hull et al., 2015).

### Extinction metrics and their statistical analysis

Extinction analyses implicitly assume that the available raw data faithfully reflect the phenomenon of biological interest. For modern species, data are written dated records of last sightings or other conservation logs – whereas the raw data for ancient taxa resides fully in the fossil record. Unfortunately, a lack of direct observations of a species occurrence (because of physical impossibility or cost) does not warrant declaring a species extinct. Similarly, muddled data acquisition due to nonhomogeneous geographic, stratigraphic or temporal sampling efforts, taxonomic errors, and dating errors, among others, can lead to erroneous conclusions (Foote, 2000; Alroy, 2010). Disentangling a biological conclusion from data that is further made noisy by reasons other than biology is a statistical challenge in itself that has bearing on uncertainty in declaring species extinct and in estimating rates of extinction (see Sprott, 2000, Chapter 4).

Once available data are deemed to accurately reveal the status of a species, the question arises as to how to quantify the likelihood that a species is extinct. One approach has been to quantify diversity or richness (Alroy et al., 2008) at a reference time for comparison with the number of taxa that existed over a specified time interval. This can be used to calculate the number of E/MSY (Pimm et al., 1995). More elaborate rates of extinction have been proposed (Foote, 2000; Alroy, 2010) that incorporate notions of species origin as well as extinction (see Foote, 2000, for detailed discussion of their sensitivity to potential intrinsic factors such as preservation probability and interval size).

An alternative means to quantify the *risk* of extinction instead of using numbers or proportions becoming extinct (by means of the statistical theory of survival analysis; Kleinbaum and Klein, 2005) involves characterizing the complete distribution (not only calculating its mean) of lifetimes in terms of hazard functions that are related to the probability of a taxon going extinct being conditional on its

survival up to a given time (Doran et al., 2006; Drake, 2006; Naka-mura et al., 2013). This approach to estimating extinction risk might be conveniently expanded by considering regression-type models to investigate nonhomogeneous relationships between hazard and other concurrent variables (i.e., Cox models, Kleimbaum and Klein, 2005, Chapter 3, Doran et al., 2006; Pérez-Sosa et al., 2023).

Metrics such as E/MSY calculated across two extremely different timescales (millions of years for fossils, and decades for modern species) are not readily useful for determining whether a sixth mass extinction is underway. For one thing, the different metrics have tended to be computed by counting different taxonomic levels. In addition, extinctions in the fossil era have already occurred, whereas the sixth extinction may be an ongoing process. However, there is an alternative per capita metric (applied at the genus level) that appears to overcome this limitation by explicitly considering observation timescales reduced to common metrics used at either of two extremes: geological scale versus modern scales (Spalding and Hull, 2021). It incorporates the concept of extinction debt (Kuussaari et al., 2009) and a model for mass extinctions based on stochastic pulses to compare ancient and recent extinction rates. Problems posed by comparing the past and present have been recognized for some time (Jablonski, 1994; Barnoski, 2011; Payne, 2016). However, comparing genera metrics with species metrics at contrasting time scales remains a problem with many subtleties regarding methodology and working assumptions.

One means to address the incomplete fossil record and its associated biases is to apply a technique known as rarefaction. This has been successfully applied to paleobiological data to artificially balance out samples of unequal representation and make them comparable. Such an approach represents sampling effort by observed size (Alroy, 2010) or by the completeness of species accumulation curves if there is sufficient data structure (Chao and Jost, 2012). Sampling effort can also be quantified from external, concomitant sources such as geological or rock accessibility based on correlations between fossil and rock records (Peters, 2005), although others authors believe the rock record is biased (Smith, 2007; Benton, 2009).

Uncertainty in dating is another intrinsic limitation of fossil data (Signor and Lipps, 1982) because the start and end dates of each genus range (represented by its first and last organism) may not have been preserved. Despite this limitation, the fossil record has been a valuable data source and statistical methodologies have been developed to quantify uncertainties (Strauss and Sadler, 1989; Solow, 1993). However, problems remain for species described as singletons whose genera's first and last appearance occurred in the same time interval (Hammer and Harper, 2006). Singleton taxa have tended to be ignored (e.g., Spalding and Hull, 2021), but may yet provide valuable information (Fitzgerald and Carlson, 2006).

Another suspected bias associated with fossils is the so-called "pull of the recent" effect, an apparent increase in the diversity of the fossil record toward the recent due to favorable sampling of recent deposits. However, the increase in biodiversity has been shown to be a genuine biological pattern and not an artifact of a sampling bias (Jablonski et al., 2003). Rigorous quantification of statistical uncertainty under all these conditions is needed, but is not always present in most studies that evaluate extinctions.

## Future perspectives in the study of marine extinctions

"Extinction debt" is a conceptual model that has been validated for terrestrial species (Kuussaari et al., 2009; Figueiredo et al., 2019), and

could be used to explain why marine populations are now disappearing at faster rates than did the loss of only two dozen species over the past half millennium. It could be used to gain insight into the biological and ecological processes involved, and how they determine the shortened time between initial perturbations of a population (e.g., collapse induced by overexploitation or severe habitat loss) and the extinction of a species. Such an analysis requires making sensible extrapolations of unseen extirpations, and estimating the risk of extinction of marine species in poorly studied taxa (Ricketts et al., 2005; Webb and Mindel, 2015; Pacoureau et al., 2021). Another promising approach to study marine extinctions is to determine species–area relationships as a function of the amount of marine habitat that is being lost, as shown by the linkage between regional extinction and sediment truncation in North America (Heim and Peters, 2011). This approach has been recently revisited by Spalding and Hull (2021) to derive a sedimentary proxy of extinction debt.

Determining when a population or species has gone extinct is particularly difficult to do for marine organisms because of the vast tridimensional space they inhabit, and the dissimilar ontological stages they exhibit associated with different habitats and trophic levels (del Monte-Luna et al., 2007). The International Union for the Conservation of Nature (IUCN) (2019) has adopted new methodologies to evaluate extinctions that incorporate qualitative and quantitative approaches (Akçakaya et al., 2017; Keith et al., 2017; Thompson et al., 2017). However, a lack of transparency of some of the new methods can impede making immediate decisions, as can idiosyncratic differences among some expert opinions. Thus, there is a need to continue developing new approaches to assess extirpations and extinctions of marine species that are statistically sound, easy to apply, efficient under data-poor situations, and readily applicable in tandem with existing methods.

New approaches for assessing marine extinctions will help fill at least two information gaps. The first is criteria systematization needed to determine when a population or species can be considered extirpated or extinct. The second is a means to quantify biodiversity loss in terms of the number of populations and species. The new approaches need to carefully consider the input data that will be used to estimate current extinction rates in the marine realm. For example, they should factor in some measurement or proxy of sampling effort (applied at both modern and ancient time scales). They also need to address the problem of contrasting orders of magnitude in the time dimension, as well as the issue of applying metrics at differing taxonomic levels. Only then can sensible comparisons be made between modern extinction rates and those from the distant past. Until this is addressed, it would be prudent to withhold claiming a mass extinction is underway in the marine realm.

Nakamura et al. (2013) concluded that 20 documented cases of extinction were insufficient to reliably determine that the relative extinction rate of marine species has increased or remained constant over the past 500 years. However, a few "extra" cases over the coming years could alter the statistical significance of their estimated extinction trend. Should the verdict of unresolved cases of species disappearances (Table 1) lean toward "extinction," the extinction rate of marine species would show a statistically significantly increasing trend.

In conclusion, we envision three approaches to better understand the potential and true magnitude of biodiversity loss in the world's oceans. These include: 1) estimating the extinction debt; 2) developing new methodologies to reliably determine when a marine population or species has gone extinct; and 3) improving analytical procedures to estimate rates of loss using standardized metrics that allow comparisons to be made among different regions

and taxonomic groups, and between past and current times. Quantifying how many populations and species might go extinct in the future, generating global extinction metrics and determining how they change over time will contribute to defining quantitative reference points and focusing efforts to assess the loss of marine biodiversity.

**Open peer review.** To view the open peer review materials for this article, please visit http://doi.org/10.1017/ext.2023.11.

**Data availability statement.** Data considered in this manuscript are obtained from the reference list presented below.

**Acknowledgments.** P.d.M.-L. thanks the Instituto Politécnico Nacional–IPN and its scholarships from the Estímulos al Desempeño de los Investigadores–EDI and from the Comisión de Operación y Fomento de Actividades Académicas–COFAA.

**Author contribution.** Conceptualization: P.d.M.-L.; Data curation: A.V., A.Y.-A.; Formal analysis: P.d.M.-L.; Investigation: P.d.M.-L., M.N., A.V., L.P.-S., A.Y.-A., A.W.T., S.L.-C.; Project administration: P.d.M.-L.; Validation: M.N., A.V., L.P.-S.; Visualization: A.V., A.Y.-A.; Writing-original draft: P.d.M.-L., M.N., A.V., L.P.-S., A.Y.-A., A.W.T., S.L.-C.; Writing-review and editing: P.d.M.-L., M.N., A.V., L.P.-S., A.Y.-A., A.W.T., S.L.-C.

**Financial support.** We thank Project Ciencia Básica A1S19598 of the Consejo Nacional de Ciencia y Tecnología (CONACyT) and Project SIP 20231624 of the Instituto Politécnico Nacional.

**Competing interest.** The authors declare no competing interest.

**Ethics standard.** *Authors' management* of the data and scholarship has been used objectively and without any bias. The authors confirm that the manuscript is original and has not been previously submitted to another journal.

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
