## [Reviewer Report]

*Comments to Author*: Summary

This study is presented as a review paper. Its objective is 1) provide an overview of modern marine extinctions, 2) address key concepts and methodologies that makes it difficult to contextualize and interpret the current biodiversity loss in marine ecosystems, and 3) identify priority areas for research to better quantify and contextualize current marine extinction rates.

When providing an overview of the state-of-knowledge on modern marine extinctions, the updated overview made by the authors is based on an unpublished (not peer-review) paper from Yañez-Arenas et al in prep, which can be problematic, since it cannot not yet considered “available evidence”. The authors also strongly claim multiple times in the review paper that the idea of an ongoing sixth mass extinction in the marine real should be put on hold, yet, I find this claim and their arguments put forward not well justified and not summarized well in the review paper, since in the current published literature there is a wide range of evidence and wide range of arguments debating and contextualizing the extinction rates in the marine real, which are not well captured in this review paper. I think this point by itself needs to be reconsidered before this paper is considered for publication. Specially because this paper is presented as a review article, which is supposed to be a general summary of the existing literature on this topic and an attempt to explain the current state of understand on the topic tacked.

I would also suggest removing the reference (Yañez-Arenas et al. in prep) from this review and also Table 1 since this study is still in preparation and it has not yet been peer-reviewed and published. Specially since it claims the number of declared marine extinct species is 19 compared with the most recent study (UICN 2022) which claims the number of declared marine extinct species is 27. These discrepancy in number should be reviewed, explained, and justified before it is summarized in this Table 1 and used as evidence for understanding current extinction rates in the marine realm.

However, I think the authors summarize well and identify well key important research challenges in estimating and interpreting estimations of past and present marine extinctions, for example, challenges arising when contrasting time scales and the different taxonomic levels used in the past and present calculations of extinction rates. Authors also make the effort to summarize the state of knowledge and put forward potential solutions on some of the identified challenges (how to adapt extinction risk metrics to differing time scales, or how to account for sampling effort bias), and also summarize where advances and research is still lacking (for example, how compare metrics between two periods based on different taxonomic levels is still challenge, among others).

The authors also highlight the need to spend more resources to develop robust methods for determining when a species or population has gone extinct (referred as extinction assessments) and for determining current extinction rates using standardized methods to better contextualize ongoing marine extinctions with past mass extinction events.

Other minor comments:

Table 1 nicely summarizes in a list the number of declared marine species gone extinct between 1975 and 2022 (from algae to mammals), and which species have later been declared non-extinct by some other author. Although the Table is informative, the authors could consider modifying the table to make it more visual for the general readers. For example, it could include the common name, in addition to the Latin name, and also order/group the species by taxa groups or phyla, to get a quicker idea of how extinction risk is spread across different taxa.

Page 3 – Line 49-50 – The written statement for objective number two of this MS is not clear. Consider revising the wording “we consider giving nuance” and use alternative language to clarify the aim of objective number 2.

Page 4 – Line 113. While the authors define the term of extinction and extirpation in page 3, the authors did not define the term eradication, which is only used in this line of the MS. I think the authors refer to 60 to 112 documented extirpations around the world´s ocean. Please revise or define this term.

Page 4 -Line 115-116 – Revise the sentence claiming “¾ of all extant elasmobranchs might be committed to extinction” citing Pacoureau et al 2019 paper. Most recent paper, Dulvy et al 2021 Current Biology claims 1/3 of chondrichthyans are estimated to be threatened with extinction.

Page 4 – Line 116-117 claims the number of declared extinctions is between 19 and 30 depending in what is considered “sufficient evidence”. This needs to be cited well. Authors could cite table 1, but a closer look to table 1 reveals that the lower bound of 19 species is backed up by an MS in prep (Yañez-Arenas) and the upper bound of 27 by a published study (IUCN 2022). I suggest Yañez-Arenas et al MS in prep should be peer-review and published before it is included in this paper in order to justify a declared number of extinct marine species of 19. So either Authors should wait for the Yañez paper to be published in order to complete this table (and this study) or alternatively keep the number of declared marine extinctions to 29 as commonly accepted now.

Page 4 Line 130-131 – Again, the claim of 19 declared extinct species is based on an unpublished study. This should be revised and removed from this current study until the work is prep is peer-reviewed and published.

Page 6- Line 203-204 – This sentence would benefit from presenting and explaining the alternative metric and its strengths from the most current accepted metric (E/MSY). And I find Line 205-208 a bit confusing, and I suggest rewriting it to clarity its message and point, or showing an example of this survival analysis and how it is used to quantify the risk of extinction, and how this is a potential alternative and its novelty.

Page 8 – Line 294-295 – I agree that the application of IUCN methodologies to estimate extinction risk has been more applied to terrestrial populations than marine populations historically, but the number of IUCN assessment of marine species has increased remarkably in the last decade and should be acknowledged.

Page 8 – Line 295 -296- I find the discussion confusing. The word “therefore” perhaps is not needed to connect the two sentences. Or the sentence claiming that there is a need for new approaches for assessing extirpations and extinction of marine species is not well justified, based on the sentence above it.

Page 8 – Line300-313 – I agree with the authors that there is a need for 1) better determining when a population or species can be considered extirpated or extinct (referred as criteria systematization), 2) for better estimates of current extinction rates in the marine reals, 3) better ways and metrics to estimate past and current extinctions. Yet, these current challenges and needs, should not be used as evidence to claim, or not claim, whether there is an ongoing mass extinction in the marine realm. There are other lines of evidence and well supported arguments in this study and other published studies (e.g. Luypaert et al 2020, Webb et al 2015, Dulvy et al 2021) that do not fully support the claim that the extinction rate in the marine realm are lower than in the terrestrial real and that should be put on hold.

---

## [Reviewer Report]

*Comments to Author*: The review manuscript titled “Recent and future marine extinctions” by del Monte-Luna et al. tackles a timely and important issue that is inline with the journalʻs mission. In this study, the authors review the available evidence for extinction of marine species in the modern biodiversity crisis and discuss the difficulties of putting these extinctions into their proper historical context in the geologic record. The study finds that somewhere between 19-30 marine species have documented extinctions and nicely discusses and reviews why there is this range of uncertainty. The authors discuss some common metrics for estimating extinction rates and why it is difficult to make “apples to apples” comparisons between the modern extinction rate and extinction rates in the fossil record. The manuscript has many strengths and I believe may ultimately help push the field in a positive direction, however I believe that a few key pieces are missing from this review and that some of what is covered is incomplete. If these major issues can be adequately addressed, I believe this manuscript has the potential to be published in Cambridge Prisms: Extinctions and make an excellent contribution to the field. I detail these major issues below and will reserve the more minor issues for now.

Major issues

1. The paleo component of the review is weakly developed. While the papers cited are a good start for the discussion in the field of how (including the many inherent challenges) to compare the modern biodiversity crisis to both prior background and mass extinction intervals, many key publications are completely missing. For example, there is not any discussion of the declining extinction rate of marine species across the Phanerozoic (nor citation of Raup and Sepkoski Science 1982 – see also Benton Science 1995 for terrestrial species that show the same decline in extinction rates across time) and what this means for comparison to the modern rate and putting it into proper context (that is the rate is changing across time in a predictable way). Also, see Knope et al Science 2020 for a mechanistic ecological explanation for this decline in extinction rates over evolutionary time. Also a few other key papers for comparison of marine extinction in the modern to the fossil record are missing (among others):

Finnegan S et al., Science 2015

Payne JL et al., Science 2016

Payne JL et al., Biology Letters 2016

2. Extinction events (modern or ancient) are generally evaluated on three criteria: 1) rate; 2) magnitude; and 3) selectivity. This review toggles between evaluation of rate and magnitude, without clearly and convincingly connecting the two - and perhaps more importantly, ignores selectivity altogether. If the authors want to make an argument for ignoring the patterns of extinction selectivity in the modern and how it compares to the fossil record, they should be explicit about why they consider it to not be important in a review on recent and future marine extinctions. Here are a few key studies on extinction and extinction risk selectivity in the fossil record and the modern as a starting point:

Knoll AH et al., EPSL 2007

Janevski GA and TK Baumiller Paleobiology 2009

Atwood T et al., Science Advances 2020

Munstermann M et al., Conservation Biology 2020

3. While the title of the manuscript includes “future marine extinctions” it feels that little attention is paid to extinction debt and what the trajectories of so many marine species with rapidly declining populations throughout their range are - given marine ecosystem management continues in a “business as usual” manner. If greater attention could be paid to the depopulation trends and to functional extinctions - I believe the manuscript would better portray the actual state of the crisis in the oceans. On this note, I donʻt think functional extinction is brought up at all in the study, when it is likely the more important issue in the modern crisis than full extinctions at this point in time – that is, extirpating every last individual of a species is required for it to be officially considered extinct, but the species stops contributing ecologically (and functionally) in a meaningful way long before it is driven entirely extinct. Further, large numbers of functional extinctions may lead to cascading effects on other species, including complete extinctions. These papers are already cited in the manuscript, but Dirzo et al. Science 2015 and MaCauley et al Science 2015 are good starting points for developing this component of the manuscript, beyond how they are cited now.

4. There is no recognition that in the modern the 19-30 documented complete extinctions of marine organisms are of only macro-organisms (macro-algae to mammals). To be more taxonomically inclusive (and to bring our microbial colleagues into the fold), I suggest that you also consider discussing what the rate, magnitude, and selectivity patterns of micro-organisms (e.g., single-celled protists, bacteria, archaea, viruses, fungi) in the oceans might be. Of course, the extinction patterns of these taxa are not well-constrained in the modern or in the fossil record, but that is exactly the point to make after explaining what we do know already – we need to work towards their inclusion if we are to fully understand the current biodiversity crisis and its proper geologic context. Of course, these microbial trajectories are tightly linked to biogeochemical cycles and ecosystem function that can impact macro-organisms in a wide variety of ways. Tangentially related, I think the manuscript could benefit from some figures – one idea would be a few simple pie charts about the taxonomic distribution of the documented extinctions (e.g., N number of extinctions in class and/or phylum XYZ – maybe one for the 19 and another for the 30?). These types of simple figures could then be incorporated into the discussion of taxonomic biases in the study of extinction. Another thought is to include a timeline figure of the dates of proposed extinctions and re-discoveries from table 1.

---

## [Reviewer Report]

*Comments to Author*: The manuscript titled “Recent and future marine extinctions” reviews the current status of the “Sixth Mass Extinction” in the oceans. In particular, it explores the apparent disconnect between the claim that oceans are experiencing a mass extinction and the very small number of documented marine extinctions. This is, of course, an important topic and one that is of wide interest to extinction scholars and the general public. To this end, I found Table 1 very interesting and informative. One major improvement for Table 1 would be to add higher taxonomy. Including Kingdom, Phylum, and Class to the table would give readers a much more intuitive sense of how documented extinctions are spread across the tree of life.

However, I found the framing of the manuscript to be a bit off the mark and content of the review too shallow for a broad audience of paleontological and neontological extinction scientists. I think the most useful review papers (line 53 indicates this is a review paper) provide a detailed description of the state of the art and lay out a roadmap for future directions. The current manuscript presents an interesting and important problem but does not explain the state of the art with any detail. Methods from specific studies (e.g., Spalding & Hull) are both praised and criticized in general terms, but they are not explained. While the manuscript recommends that science do a better job at documenting species extinction and come up with better metrics for extinction rates observed across a range of timescales, this is not what I would consider to be a specific roadmap for moving forward.

To me, the framing of the manuscript is around how previous studies of marine extinction have claimed that we are currently in a mass extinction without recognizing that the number of documented extinctions is very small (lines134-136; 312-313). Rather, these studies have used populations declines, extirpations, and other measures of extinction risk in still extant species as a proxy for extinction. However, I find this to be a mischaracterization of the literature. Previous studies that use population declines etc. do indeed recognize and state explicitly that a marine mass extinction has not yet been realized, but if at-risk species continue to have population declines that restful in extinction, then the oceans will experience a mass extinction. To me, this is entirely reasonable and much more informative than only looking at documented extinctions and concluding there is no extinction event (lines 312-313). Of course, the manuscript is not claiming that is nothing going on viz-a-viz extinction, but the criticism of the existing research is unwarranted. In fact, given the small number of documented extinctions, monitoring extinction risk and making predictions about the future is more productive than waiting for more species to be documented as already extinct.

I found the treatments of most topics in the manuscript rather shallow for a review paper targeted at a broad audience. I think a much deeper exploration of why there are so few documented marine extinction would be quite informative. For example, the 20-30 documented extinctions could be the true number of extinctions over the past 500 years, but it could also be artificially low because of under-study. A quick scan of the IUCN Red List suggests that a small proportion of marine species have been evaluated—at least in comparison to terrestrial species. A discussion of why and how the oceans are understudied would be instructive. Relatedly, several studies have suggested, I think correctly, that marine species have been buffered from anthropogenic disturbances. While coastal resources have been exploited since the end of the Pleistocene on small scales, large impacts did not materialize until the industrial age which ushered in changed in terrigenous sediment delivery, pollution, and industrialized fishing in the open ocean.

One of the manuscript’s main take aways is that new metrics for measuring extinction need to be developed. However, several new metrics are praised (lines 203-208) without either explaining in detail what they are or explaining how they fall short (as the call for new metrics implies). Moreover, given the very small number of documented extinctions, I am not sure that a new metric will be any more informative than continuing to quantify and track at-risk populations.

I found the lack discussion of Payne et al. (2016) to be an oversight (in full disclosure, I’m a co-author on that paper). While Payne et al. is focused on extinction selectivity, it does address most of the issues raised in the manuscript with regards to extinction magnitude. It makes an “apples-to-apples” comparison of the fossil and recent records. And while it uses extinction threat + documented extinctions rather than exclusively relying on documented extinctions, it generates a range of potential extinction rates that range from background to mass extinction.

The discussion of the fossil record’s biases is also too shallow and lacking nuance. I worry that readers who are not deeply familiar with the structure of the geologic and fossil records will be left with the impression that they are biased to the point of not being useful. For example, the point is made that last appearances in the fossil record are rarely true last appearances (line 233). While this is true, there has been considerable work by Sadler, Marshall, Holland, and others to construct confidence intervals on observed last occurrences. The manuscript also claims that quantity of sedimentary rocks of different ages is an unexplored problem (line 229) without discussion of the work by Raup, Peters, Smith, Dunhill and others on this topic.

Two more minor points:

I found the manuscript to be lacking in several appropriate citations (e.g., for “Big 5” Mass extinctions, Signor-Lipps Effect, Pull of the Recent).

There are several uses of words that attribute value to scientific data and results. For example, “optimistic” (line 132) and “dire” (line 265). While I personally agree with the authors that anthropogenically-driven extinctions are bad (and I suspect most readers would too), these terms are in fact ambiguous. E.g., “Using the upper estimate of extinction” is much more clear than “Using the most dire numbers”.

---

## [Editor Report]

*Comments to Author*: I have now received reviews of this ms from three expert reviewers. All three of them find aspects of the ms to be useful and insightful, with the potential to make a real positive impact on the field, but they have also all raised concerns that are sufficiently major that some substantial changes are required. I would encourage the authors to carefully consider all three reviews, as there are a number of interesting ideas and constructive suggestions in all of them. However, I do recognise that several of the concerns are challenging to address within the constraints of the article type - Review articles are quite a short (3000-4000 word) format. This means that the constructive ideas for adding more depth that the reviewers provide may not be feasible and the final ms may need to remain more ‘broad brush’. I would encourage the authors to consider if they could reasonably include more detail of, for example, specific methods for estimating extinction (Reviewer 1) or why there are so few documented marine extinctions (Reviewer 1) - is this simply a data gap? It has been shown for example that IUCN assessments are concentrated within the taxonomically best known marine groups (see Webb & Mindel 2015 https://doi.org/10.1016/j.cub.2014.12.023) - does that in part explain the lack of observed extinctions? Or is it (as Reviewer 1 suggests) that marine species have so far been less exposed to the human activities driving extinctions (this is the argument made in McCauley in Marine Defaunation). I suspect a mix of the two is closest to the truth. More clarity on the distinctions between extinction rate, magnitude, and selectivity is also needed (Reviewer 2).

It may not be possible within the available space to fully cover issues of extinction in the fossil record (Reviewers 1 and 2), however more thorough citation of the relevant literature may help to address this point - both these reviewers provide some useful suggestions. Reviewer 2 makes an interesting point about microbial extinction rates - clearly microbes are often overlooked but play essential roles within ecosystems. However I do not feel it would be possible to do the topic justice within this review - but maybe making explicit mention of this omission would be sensible.

Even given the constraints of the ms format, I share the reviewers’ concerns that the literature that is reviewed may not be fairly characterised - Reviewer 3 provides some specific examples of this. All reviewers make the point that in marine systems, population declines, extinction risk, functional extinction, etc. are usually all we have to go on - and that although these are not direct measures of extinction, they should not be discounted. Perhaps a more balanced tone in discussing the published evidence would be appropriate. The Review format is, as the name suggests, intended to present a review on the current state of knowledge in a major subject field. This ms reads more like an opinion piece in places - I think a degree of subjective interpretation is appropriate but not too much. A Review should also not present original research, but as Reviewer 3 notes, some of the key points presented rely on a currently unpublished study (listed as in prep) and using this to counter other, formally published estimates is rather problematic - especially as the unpublished estimate sets the lower bound for the range of estimates listed in table 1.

As a final point - all reviewers make suggestions for how the visual components of the ms could be improved, through better design of the Table and/or inclusion of a figure. I agree that this could really increase the impact of the work and I would encourage the authors to give this some consideration.

---

## [Reviewer Report]

*Comments to Author*: This is my second review of the manuscript titled “A Review of recent and future marine extinctions”. Overall, the manuscript is improved, but I have a few lingering comments.

Table 1 is much improved! I think this will be useful to extinction scholars.

My major overarching concern is that the manuscript is unbalanced with regards to the challenges of comparing fossil and recent extinction metrics. The section titled “Extinction metrics and their statistical analysis” is devoted almost entirely to the problems of the fossil record. The second sentence of that section states that modern extinction records are derived from “written dated records of last sightings or other conservation logs”. However, the remainder of the paragraph and the next six paragraphs discuss only problems with measuring extinction in the fossil record. Are there not problems with recognizing extinctions in the modern? Are records from Victorian era European/American explorers a true random sample of marine biodiversity? Are earlier records made by European colonizers not without bias and hyperbole? Is the geographic coverage of species threat assessments uniform or is it highly idiosyncratic? Is it not true that oceanographers still make the claim that we know more about the moon than we do the ocean? It is also obvious from a quick scan of the Red List that there is a very strong bias in the evaluations of marine species that skews heavily towards charismatic megafauna and commercially important fish and invertebrates. (As an aside, an important caveat of this observation is that most information we have from the fossil record are from decidedly uncharismatic invertebrates.) All scientific data is imperfect and incomplete, thus it seems prudent to critically evaluate the nature of all the data needed to answer the question at hand.

Despite my comment above and given the word limit for the manuscript, I am not convinced that quite so much text should be devoted to the details of the nature of the paleontological and neontological records. Acknowledging that there are limitations and assumptions is a good idea, but unless there is something specific that is keeping us from answering of the focal question, are the oceans currently experiencing a mass extinction?, I would keep this section brief and allow more space for more relevant aspects of the problem. The manuscript argues, I think, that the two biggest challenges are comparing extinctions across two different timescale and comparing fossil genera to recent species. I suggest devoting more text to these issues and less to the particulars of the fossil record.

Minor comments and suggestions:

Line 57: Is it necessary to put ‘terrestrial’ in parentheses? The so-called sixth extinction has been largely defined by terrestrial animals while the “big 5” mass extinctions of the fossil record have been defined by marine animals—though clear terrestrial vertebrate extinctions have also been recognized in the latter three.

Lines 153: I wouldn’t say that mass extinctions are non-selective or only selective agains small-bodied taxa. Rather, mass extinctions tend to be differently selective than background times. Some traits that provide extinction resistance in background time do not provide protection during mass extinctions. Jablonski (1986 Science) and Payne & Finnegan (2007 PNAS) found that geographic range protects against extinctions during background but not mass extinction intervals. Payne et al. (2016 Science) found that pelagic chordates and mollusks were select for extinction during all mass extinction intervals and small-bodied chordates and mollusks during some mass extinctions. During the Permian mass extinction, marine animals that were “physiologically buffered” preferentially survived (Knoll et al. 2007 EPSL).

Lines 162: This paragraph seems to be only about marine extinction risk. However, Munstermann et al. 2021 is a study of terrestrial vertebrates only.

Lines 255-256. Yes, comparing rates computed for genera vs. species is a problem. However, different researchers have attempted to resolve the issue—of course imperfectly and with assumptions. Payne et al. (2016 Science), for example “downgrades” modern taxonomic data to the genus-level for comparison with fossil rates. More commonly, information such as the genus to species ratio is used to estimate species-level extinction rates from genus-level data (Jablonski 1994 Phil. Trans. Ro. So.). Barnosky et al. (2011 Nature) does this particularly well by using estimated species-level thresholds for mass extinctions, then estimating the amount of time needed to get to that level based on Red List extinction threats. Given the challenges of correlating fossil species-level occurrences globally, some flavor of these solutions is probably the best we can do—at least for now. By ending the paragraph by stating that the problem is open, the reasonable solutions that have already been proposed and applied are ignored or dismissed.

---

## [Reviewer Report]

*Comments to Author*: I commend the authorʻs on doing a nice job of revising this manuscript to address my comments, as well as those of the other reviewers and the editors. It is very much improved from the previous version and will make a very nice contribution to the literature.

---

## [Editor Report]

*Comments to Author*: Thank you for conducting such a thorough revision of your manuscript. This has now been considered by myself and by two of the original reviewers. As you will see, one reviewer is fully satisfied with your revisions, while the other has some remaining minor concerns and suggestions. My own view is that the more ‘major’ of these suggestions, which would require more substantial rewriting and restructuring, are beyond the scope of your ms. Within the constraints of the ms format, I think you have produced a coherent and complete piece with a clear viewpoint, and while I can see that adopting the reviewer’s suggestions in full would strengthen some aspects, this would necessarily be at the expense of others. Therefore I am recommending that we accept your manuscript in its current form, as I believe it will stimulate discussion in this important area. However, I would strongly encourage you to consider if some of the more minor suggestions from reviewer 2 could be incorporated at this stage, to further improve your work.